# Quantitative Analysis of Photon Density of States for One-Dimensional Photonic Crystals in a Rectangular Waveguide

**Ruei-Fu Jao [1] and Ming-Chieh Lin [2,*]**

[1] School of Information Technology, Guangdong Industry Polytechnic, Guangdong 510300, China; 2014108048@gditc.edu.cn

[2] Multidisciplinary Computational Laboratory, Department of Electrical and Biomedical Engineering, Hanyang University, Seoul 04763, Korea

* Correspondence: mclin@hanyang.ac.kr; Tel.: +82-2-2220-0358

**Abstract:** Light propagation in one-dimensional (1D) photonic crystals (PCs) enclosed in a rectangular waveguide is investigated in order to achieve a complete photonic band gap (PBG) while avoiding the difficulty in fabricating 3D PCs. This work complements our two previous articles (Phys. Rev. E) that quantitatively analyzed omnidirectional light propagation in 1D and 2D PCs, respectively, both showing that a complete PBG cannot exist if an evanescent wave propagation is involved. Here, we present a quantitative analysis of the transmission functions, the band structures, and the photon density of states (PDOS) for both the transverse electric (TE) and transverse magnetic (TM) polarization modes of the periodic multilayer heterostructure confined in a rectangular waveguide. The PDOS of the quasi-1D photonic crystal for both the TE and TM modes are obtained, respectively. It is demonstrated that a "complete PBG" can be obtained for some frequency ranges and categorized into three types: (1) below the cutoff frequency of the fundamental TE mode, (2) within the PBG of the fundamental TE mode but below the cutoff frequency of the next higher order mode, and (3) within an overlap of the PBGs of either TE modes, TM modes, or both. These results are of general importance and relevance to the dipole radiation or spontaneous emission by an atom in quasi-1D periodic structures and may have applications in future photonic quantum technologies.

**Keywords:** photonic crystals; photonic band gap; waveguide; complete PBG; PDOS; TE; TM

---

## 1. Introduction

Photonic crystals (PCs), also known as artificial materials, have attracted much attention in the past three decades due to the tremendous needs of gaining complete control over light propagation and emission [1,2]. PCs, according to the dimension of the periodicity, are divided into three categories: one- (1D), two- (2D), and three-dimensional (3D) crystals. Due to the periodicity, the stop band and the pass band are formed, according to the Bloch theorem [3]. Therefore, periodic dielectric materials are characterized by photonic band gaps (PBGs). As an analogy to the electronic band gaps in solid state materials [4], a PBG in PCs can prohibit the propagation of electromagnetic (EM) waves whose frequencies fall within the band gap region. The continuing success of the semiconductor industry in controlling electric properties of materials from the last century has encouraged us to manipulate the flow of light in PCs and control their optical properties. These PC-based materials are expected to have many applications in optoelectronics, optical communications, and photonic quantum technologies in the next decades [5]. In optical range, PCs have been extensively studied. It was proposed that the emission of EM radiation can be modified by the environment [6,7]. Several environments such as

metallic cavities [8], dielectric cavities [9], superlattices [10–15], and 2D PCs [16] have been studied. The environmental effects have been described by the photon density of states (PDOS), which is related to the transition rate of the Fermi golden rule. In principle, a PC with a complete PBG can be best realized in a 3D system to prohibit the propagation of electromagnetic waves of any polarization traveling in any direction from any source [1]. However, the difficulty in fabricating such 3D crystals with PBGs in the optical regime prohibits the progress of many applications.

On the other hand, there has been a lot of interest in microwave and millimeter wave applications of PBG, such as the significant progress in the design of filters [17,18], microstrip antennas [19], and slow wave structures [20,21], and so on. However, the design of PBG in this frequency range is still difficult due to complexities of the modeling. There are too many parameters affecting the PBG properties, such as the number of lattice periods [22], lattice shapes [23,24], lattice spacing [25], and relative volume fraction [26–30]. Since the actual fabrication of 3D PCs remains difficult, another simpler choice is periodic dielectric or PC waveguides, which have only a one-dimensional periodic pattern [1]. The rigorous study of the PC waveguide can be traced back to the 1970s, where a more detailed review can be found [31,32]. Recently, it was demonstrated that, by considering a quantum dot spin coupled to a PC waveguide mode, the light–matter interaction can be asymmetric, leading to unidirectional emission and a deterministic entangled photon source, which might have application in future optical quantum devices [5,33]. One interesting feature of electromagnetism in dielectric media is that there is no fundamental length scale, namely the scaling properties of Maxwell's equations, i.e., the solution of problem at one length scale determines the solutions at all other length scale [1]. In a previous work [34], a multilayer dielectric window in a rectangular waveguide had been studied to achieve a wide-bandwidth transmission of a millimeter wave. A transfer matrix approach was successfully employed to discretize the dielectric function profile and the transmission functions could be calculated efficiently. In principle, the approach can be extended to study a quasi-1D PC, a PC confined in a waveguide. However, the transmission method is limited to study radiation modes in a finite-length system. In order to study the PBG phenomena such as the suppression of spontaneous emission [35] in a quasi-1D PC, the calculation of the dispersion relations or band structures (BS) and the PDOS are needed. Metallic waveguides and cavities are widely used to control microwave propagation. One of the main concerns is visible light energy is quickly dissipated within the metallic components, which makes this method of optical control almost impossible to generalize. Recently, an unconventional superconductivity in magic-angle graphene superlattices had been discovered and studied [36]. The superconductivity might help realize the metallic waveguide confinement of optics in the near future.

In this work, light propagation in 1D PCs enclosed in a rectangular waveguide or quasi-1D PCs is investigated in order to achieve a complete PBG while avoiding the difficulty in fabricating 3D PCs. This work complements two previous articles [15,16] that quantitatively analyzed omnidirectional light propagation in 1D and 2D PCs, respectively, both showing that a complete PBG cannot exist if an evanescent wave propagation is involved. The transfer matrix method is extended to study the transmittance of the quasi-1D PCs for both TE and TM polarization modes [34]. The corresponding BS are obtained by solving the eigenvalue equations with proper periodic boundary conditions following the Bloch theorem [3,4]. The formulas for evaluating the PDOS of the quasi-1D PCs for TE and TM modes are derived, respectively, for determining the PBGs. The contributions of the PDOS from each modes can be distinguished. The model is formulated in Section 2. The calculated results and discussion are presented in Section 3. The conclusions are given in Section 4.

## 2. Formulations

A transfer matrix approach is employed to discretize the dielectric function profile of the dielectric multilayer heterostructures and the transmission functions are calculated by matching the boundary conditions at each interfaces. In order to solve the PDOS, it is necessary to calculate the dispersion

relation, and the corresponding band structures are obtained by solving the eigenvalue equations with proper periodic boundary conditions.

## 2.1. Transfer Matrix Method

Let us consider a waveguide with its rectangular cross section of sides $a$ and $b$, and the enclosed multilayer dielectric slab with thickness, $(t_1, t_2, t_1, t_2, ...)$ and dielectric function, $(\varepsilon_1, \varepsilon_2, \varepsilon_1, \varepsilon_2, ...)$, as shown in Figure 1. The TE mode (H mode) and TM mode (E mode) in the rectangular waveguide are characterized by the $z$ components of the magnetic field and the electric field, $H_z$ and $E_z$, respectively. By definition, these components are never absent in the corresponding modes. The $z$ components of the Helmholtz's equations for the inhomogeneous media are

$$\left\{ \varepsilon(z) \vec{\nabla} \times \left[ \frac{1}{\varepsilon(z)} \vec{\nabla} \times \vec{H} \right] \right\}_z + \omega^2 \varepsilon(z) \mu(z) H_z = 0 \tag{1}$$

and

$$\left\{ \mu(z) \vec{\nabla} \times \left[ \frac{1}{\mu(z)} \vec{\nabla} \times \vec{E} \right] \right\}_z + \omega^2 \varepsilon(z) \mu(z) E_z = 0. \tag{2}$$

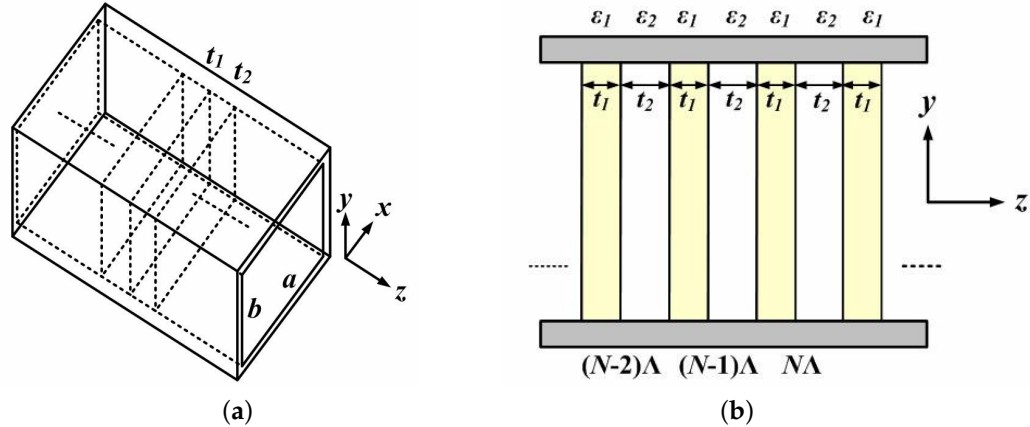

**Figure 1.** (**a**) 3D schematic of periodic multilayer heterostructure along the $z$-direction confined in a rectangular waveguide with a width $a$ and a height $b$ and (**b**) the corresponding dielectric function profile $\varepsilon(z)$ in the cross-sectional view.

In these cases, the effect of losses of the medium inside the waveguide is characterized by the complex permittivity $\varepsilon(z)$ and permeability $\mu(z)$. Thus, Equations (1) and (2) can be rearranged as:

$$\left[ \frac{\partial^2}{\partial x^2} + \frac{\partial^2}{\partial y^2} + \varepsilon(z) \frac{\partial}{\partial z} \frac{1}{\varepsilon(z)} \frac{\partial}{\partial z} \right] H_z + \omega^2 \varepsilon(z) \mu(z) H_z = 0 \tag{3}$$

and

$$\left[ \frac{\partial^2}{\partial x^2} + \frac{\partial^2}{\partial y^2} + \mu(z) \frac{\partial}{\partial z} \frac{1}{\mu(z)} \frac{\partial}{\partial z} \right] E_z + \omega^2 \varepsilon(z) \mu(z) E_z = 0. \tag{4}$$

By symmetry, using separation of variables, Equations (3) and (4) can be split into transverse and longitudinal parts, and the problem can be simplified as solving the one-dimensional Helmholtz's equation along the $z$ direction, the longitudinal parts,

$$\varepsilon(z) \frac{\partial}{\partial z} \frac{1}{\varepsilon(z)} \frac{\partial}{\partial z} \psi(z)_{TE} + \left[ \omega^2 \varepsilon(z) \mu(z) - k_c^2 \right] \psi(z)_{TE} = 0 \tag{5}$$

and

$$\mu(z) \frac{\partial}{\partial z} \frac{1}{\mu(z)} \frac{\partial}{\partial z} \psi(z)_{TM} + \left[ \omega^2 \varepsilon(z) \mu(z) - k_c^2 \right] \psi(z)_{TM} = 0, \tag{6}$$

with eigenvalues $k_c$ which are determined by the following eigenvalue equations, the transverse parts,

$$\left( \frac{\partial^2}{\partial x^2} + \frac{\partial^2}{\partial y^2} + k_c^2 \right) \phi(x,y)_{TE} = 0 \tag{7}$$

and

$$\left( \frac{\partial^2}{\partial x^2} + \frac{\partial^2}{\partial y^2} + k_c^2 \right) \phi(x,y)_{TM} = 0. \tag{8}$$

The corresponding boundary conditions for $\phi(x,y)_{TE}$ and $\phi(x,y)_{TM}$ are

$$\frac{\partial}{\partial x} \phi(x,y)_{TE}|_{x=0,a} = 0 \quad and \quad \frac{\partial}{\partial y} \phi(x,y)_{TE}|_{y=0,b} = 0, \tag{9}$$

and

$$\phi(x,y)_{TM}|_{x=0,a} = 0 \quad and \quad \phi(x,y)_{TM}|_{y=0,b} = 0. \tag{10}$$

It then follows that

$$k_c = \frac{2\pi}{\lambda_c} = \sqrt{\left(\frac{m\pi}{a}\right)^2 + \left(\frac{n\pi}{b}\right)^2}. \tag{11}$$

Applying the eigenvalue equations, Equations (7) and (8), and the boundary conditions, Equations (9) and (10), to the general solutions of Equations (1) and (2), the following particular solutions can be found by separation of variables,

$$H_z(x,y,z) = H_0 \cos\left(\frac{m\pi x}{a}\right) \cos\left(\frac{n\pi y}{b}\right) \psi(z)_{TE} \tag{12}$$

and

$$E_z(x,y,z) = E_0 \sin\left(\frac{m\pi x}{a}\right) \sin\left(\frac{n\pi y}{b}\right) \psi(z)_{TM}, \tag{13}$$

where $H_0$ and $E_0$ are determined by the energy of electromagnetic waves propagating inside the waveguide, and $m$ and $n$ are integers. The function $\psi(z)$ is chosen for a particular solution since it represents propagating waves in the $z$-direction. A transfer matrix approach is employed to discretize the dielectric function profile of the heterostructure. For an $N$-layer dielectric-filled waveguide, $\varepsilon(z)$ and $\mu(z)$ can be divided into $p = 1, 2, ..., (N+2)$ layers with a piecewise constant permittivity $\varepsilon_p$ and constant permeability $\mu_p$, respectively. The discretized one-dimensional Helmholtz's equation, for the $p$th region with constant permittivity $\varepsilon_p$ and constant permeability $\mu_p$ can be written as

$$\frac{d^2}{dz^2} \psi_p(z) + k_p^2 \psi_p(z) = 0 \text{ for } z_{p-1} \leq z \leq z_p \tag{14}$$

with

$$k_p = \sqrt{\omega^2 \varepsilon_p \mu_p - k_c^2} = \frac{2\pi}{\lambda} \sqrt{\varepsilon_{rp}\mu_{rp} - \left(\frac{\lambda}{\lambda_c}\right)^2}, \tag{15}$$

where $\psi_p(z)$ represents the wave function in the $p$th layer, and $k_p$ defines the complex wavevector in the same layer along the $z$-direction, $\lambda$ represents the wavelength in free space at the operating angular frequency $\omega$, $\varepsilon_{rp}$ and $\mu_{rp}$ are the relative dielectric constant and permeability of the medium, respectively, and $\lambda_c$ is the cutoff wavelength. The solutions of Equations (12) and (13) can be written as a superposition of the forward and backward traveling wave functions:

$$\psi_p = a_p \exp(-jk_p z) + b_p \exp(jk_p z) \quad for \quad z_{p-1} \leq z \leq z_p. \tag{16}$$

The boundary conditions for $\psi(z)$ at the interface between layers $p$ and $(p+1)$ at position $z = z_p$ where $p = 1, 2, ..., (N+1)$ are (for TE modes)

$$\mu_p \psi_p(z_p) = \mu_{p+1} \psi_{p+1}(z_p) \quad and \quad \frac{d}{dz}\psi_p(z_p) = \frac{d}{dz}\psi_{p+1}(z_p), \tag{17}$$

and (for TM modes)

$$\varepsilon_p \psi_p(z_p) = \varepsilon_{p+1} \psi_{p+1}(z_p) \quad and \quad \frac{d}{dz}\psi_p(z_p) = \frac{d}{dz}\psi_{p+1}(z_p). \tag{18}$$

By matching the boundary conditions at each discontinuity, we arrive at

$$\begin{pmatrix} a_{N+2} \\ b_{N+2} \end{pmatrix} = M_{N+1} \cdots M_p \cdots M_1 \begin{pmatrix} a_1 \\ b_1 \end{pmatrix} = \begin{pmatrix} M_{11} & M_{12} \\ M_{21} & M_{12} \end{pmatrix} \begin{pmatrix} a_1 \\ b_1 \end{pmatrix}, \tag{19}$$

where for TE modes:

$$M_p = \frac{1}{2} \begin{bmatrix} \exp(jk_{p+1}z_p) & 0 \\ 0 & \exp(-jk_{p+1}z_p) \end{bmatrix} \cdot \begin{pmatrix} \frac{\mu_p}{\mu_{p+1}} + \frac{k_p}{k_{p+1}} & \frac{\mu_p}{\mu_{p+1}} - \frac{k_p}{k_{p+1}} \\ \frac{\mu_p}{\mu_{p+1}} - \frac{k_p}{k_{p+1}} & \frac{\mu_p}{\mu_{p+1}} + \frac{k_p}{k_{p+1}} \end{pmatrix} \cdot$$
$$\begin{bmatrix} \exp(-jk_{p+1}z_p) & 0 \\ 0 & \exp(jk_{p+1}z_p) \end{bmatrix}, \tag{20}$$

and for TM modes:

$$M_p = \frac{1}{2} \begin{bmatrix} \exp(jk_{p+1}z_p) & 0 \\ 0 & \exp(-jk_{p+1}z_p) \end{bmatrix} \cdot \begin{pmatrix} \frac{\varepsilon_p}{\varepsilon_{p+1}} + \frac{k_p}{k_{p+1}} & \frac{\varepsilon_p}{\varepsilon_{p+1}} - \frac{k_p}{k_{p+1}} \\ \frac{\varepsilon_p}{\varepsilon_{p+1}} - \frac{k_p}{k_{p+1}} & \frac{\varepsilon_p}{\varepsilon_{p+1}} + \frac{k_p}{k_{p+1}} \end{pmatrix} \cdot$$
$$\begin{bmatrix} \exp(-jk_{p+1}z_p) & 0 \\ 0 & \exp(jk_{p+1}z_p) \end{bmatrix}. \tag{21}$$

Using Equation (19) with $a_1 = 1$, $b_1 = r$, $a_{N+2} = t$, and $b_{N+2} = 0$, the reflection and transmission amplitudes, $r$ and $t$, can be obtained, respectively, by

$$r = -\frac{M_{21}}{M_{22}} \tag{22}$$

and

$$t = \frac{M_{11} \cdot M_{22} - M_{12} \cdot M_{21}}{M_{22}}. \tag{23}$$

The reflection and transmission coefficients, $R$ and $T$, can be implicitly represented by $S$-parameters, $S_{11}(dB)$ and $S_{12}(dB)$, as a function of the operating frequency, , for TE modes

$$S_{11}(dB) = 10log\left|\frac{M_{21}}{M_{22}}\right|^2 \quad and \quad S_{12}(dB) = 10log\frac{\mu_1 k_{N+2}}{\mu_{N+2}k_1}\left|\frac{M_{11} \cdot M_{22} - M_{12} \cdot M_{21}}{M_{22}}\right|^2, \tag{24}$$

and for TM modes

$$S_{11}(dB) = 10log\left|\frac{M_{21}}{M_{22}}\right|^2 \quad and \quad S_{12}(dB) = 10log\frac{\varepsilon_1 k_{N+2}}{\varepsilon_{N+2}k_1}\left|\frac{M_{11} \cdot M_{22} - M_{12} \cdot M_{21}}{M_{22}}\right|^2. \tag{25}$$

*2.2. Band Structures*

In this model, $E_z$ and $H_z$ both are periodic with period $\Lambda$. According to the Bloch theorem, the electric and magnetic fields in a periodic layered medium are $E_z(z) = E_z(z + \Lambda)$ and $H_z(z) = H_z(z + \Lambda)$, respectively. The column vector of the Bloch wave satisfies the following eigenvalue equation for consistency

$$\begin{pmatrix} a_3 \\ b_3 \end{pmatrix} = \frac{1}{4} \begin{pmatrix} A_{TE/TM} & B_{TE/TM} \\ C_{TE/TM} & D_{TE/TM} \end{pmatrix} \cdot \begin{pmatrix} a_1 \\ b_1 \end{pmatrix} = e^{jk_B\Lambda} \begin{pmatrix} a_1 \\ b_1 \end{pmatrix}. \tag{26}$$

For the TE mode:

$$A_{TE} = e^{j(k_1 t_1 - k_2 t_2)} \left( \frac{\mu_1}{\mu_2} - \frac{k_1}{k_2} \right) \left( \frac{\mu_2}{\mu_1} - \frac{k_2}{k_1} \right) +$$
$$e^{-j(k_1 t_1 + k_2 t_2)} \left( \frac{\mu_1}{\mu_2} + \frac{k_1}{k_2} \right) \left( \frac{\mu_2}{\mu_1} + \frac{k_2}{k_1} \right), \tag{27}$$

$$B_{TE} = e^{-j[k_1 t_1 - k_2(2t_1 + t_2)]} \left( \frac{\mu_1}{\mu_2} + \frac{k_1}{k_2} \right) \left( \frac{\mu_2}{\mu_1} - \frac{k_2}{k_1} \right) +$$
$$e^{j[k_1 t_1 + k_2(2t_1 + t_2)]} \left( \frac{\mu_1}{\mu_2} - \frac{k_1}{k_2} \right) \left( \frac{\mu_2}{\mu_1} + \frac{k_2}{k_1} \right), \tag{28}$$

$$C_{TE} = B_{TE}^*, \tag{29}$$

and

$$D_{TE} = A_{TE}^*. \tag{30}$$

For the TM mode:

$$A_{TM} = e^{j(k_1 t_1 - k_2 t_2)} \left( \frac{\varepsilon_1}{\varepsilon_2} - \frac{k_1}{k_2} \right) \left( \frac{\varepsilon_2}{\varepsilon_1} - \frac{k_2}{k_1} \right) +$$
$$e^{-j(k_1 t_1 + k_2 t_2)} \left( \frac{\varepsilon_1}{\varepsilon_2} + \frac{k_1}{k_2} \right) \left( \frac{\varepsilon_2}{\varepsilon_1} + \frac{k_2}{k_1} \right), \tag{31}$$

$$B_{TM} = e^{-j[k_1 t_1 - k_2(2t_1 + t_2)]} \left( \frac{\varepsilon_1}{\varepsilon_2} + \frac{k_1}{k_2} \right) \left( \frac{\varepsilon_2}{\varepsilon_1} - \frac{k_2}{k_1} \right) +$$
$$e^{j[k_1 t_1 + k_2(2t_1 + t_2)]} \left( \frac{\varepsilon_1}{\varepsilon_2} - \frac{k_1}{k_2} \right) \left( \frac{\varepsilon_2}{\varepsilon_1} + \frac{k_2}{k_1} \right), \tag{32}$$

$$C_{TM} = B_{TM}^*, \tag{33}$$

and

$$D_{TM} = A_{TM}^*. \tag{34}$$

The phase factor $e^{jk_B\Lambda}$ is thus the eigenvalue and satisfies the secular equation

$$\begin{vmatrix} A_{TE/TM} - e^{jk_B\Lambda} & B_{TE/TM} \\ C_{TE/TM} & D_{TE/TM} - e^{jk_B\Lambda} \end{vmatrix} = 0. \tag{35}$$

Finally, the dispersion relation for the Bloch wave function is

$$k_B(k_z, \omega) = \frac{1}{t_1 + t_2} \cos^{-1}[\frac{1}{2}(A_{TE/TM} + D_{TE/TM})]. \tag{36}$$

*2.3. Photon Density of States*

The quasi-one-dimensional photonic crystal has been confined in the *xy*-plane, so the wave vectors $k_x$ and $k_y$ are determined according to the guiding modes. To perform the PDOS calculation, it is required to use the formal definition which is the number of available photon modes per unit frequency range. Then, we construct two frequencies, namely, $\omega(k_B) = \omega$ and $\omega(k_B) = \omega + \Delta\omega$, where $\Delta\omega$ is a small increment. We calculate the line therein, and divide it by the line segment occupied by a single mode. The differential line element in *K* space within the frequency range, is given by $\Delta L_k = \Delta k_B$. Now, $\Delta k_B$ is defined as

$$\Delta k_B = \frac{\Delta\omega}{|\nabla_k\omega|}. \tag{37}$$

Integrating over the frequency increment, we have that the total phase space line segment contributing to the frequency range $(\omega, \omega + d\omega)$ is

$$\int_{\omega_k} dL_k = \int_{\omega_k} \frac{1}{|\nabla_k\omega|} d\omega, \tag{38}$$

where we take the limit of infinitesimal increments. The number of modes within the range $(\omega, \omega + d\omega)$ is obtained by dividing the length calculated in Equation (38) by the line segment corresponding to one mode, $2\pi/\Lambda$ in the phase space. This yields

$$dN(\omega) = \frac{\Lambda}{2\pi} \int_{\omega_k} \frac{1}{|\nabla_k\omega|} d\omega \equiv D(\omega) d\omega. \tag{39}$$

Because $\omega$ is a function of *k* , we can write

$$\nabla_k\omega = \frac{d\omega}{dk_B}\hat{\mathbf{z}}. \tag{40}$$

For the TE mode:

$$\nabla_k\omega_{TE} = -\frac{(t_1 + t_2)\sin[0.5k_B(t_1 + t_2)]}{\alpha_1 + \alpha_2 + \alpha_3 + \alpha_4 + \alpha_5 + \alpha_6}\hat{\mathbf{z}} \tag{41}$$

and for the TM mode:

$$\nabla_k\omega_{TM} = -\frac{(t_1 + t_2)\sin[0.5k_B(t_1 + t_2)]}{\beta_1 + \beta_2 + \beta_3 + \beta_4 + \beta_5 + \beta_6}\hat{\mathbf{z}}, \tag{42}$$

where the functions, $\alpha_i$ and $\beta_i$, with $i = 1, 2, ..., 6$, are listed in the Appendix A. Finally, the formula for evaluating the PDOS can be expressed as

$$D(\omega)_{TE/TM} = \frac{\Lambda}{2\pi} \int_{\omega_k} \frac{1}{|\nabla_k\omega_{TE/TM}|} d\omega. \tag{43}$$

## 3. Results and Discussion

All of macroscopic electromagnetism, including the propagation of light in a photonic crystal, is governed by the four macroscopic Maxwell's equations with no free charges or currents. One interesting feature of electromagnetism in PCs is that there is no fundamental length scale other than the assumption that the system is macroscopic [1]. Therefore, to study physical phenomena in PCs, one may scale a system from the optical frequency range to the microwave one and vice versa if suitable conditions are fulfilled. For the purposes of demonstration and easier verification by experimentalists, a WR28 (7.11 mm × 3.555 mm) rectangular waveguide, usually used for Ka-band millimeter waves, is chosen. The periodic dielectric heterostructure is arranged along the longitudinal (*z*) direction to form the quasi-1D PCs [34]. In the microwave or millimeter wave frequency ranges, the PC experiments are very popular and more affordable compared to those in the optical frequency ones. Nevertheless, in order to extend the results for general use, the data are normalized for solutions

at all length scale. Consider the quasi-1D PCs with the arrangement shown in Figure 1, 15 double-layer stacks ($n$ = 30) have been included in the waveguide for investigation. The transmittance is calculated using the transfer matrix approach mentioned above. In order to demonstrate the stop band and pass band, the transmittances, expressed as the $S$-parameters, $S_{12}$, in dB, of the lowest TE and TM modes for three quasi-1D PCs have been calculated and plotted in Figure 2. For the three cases, the dielectric constants used are the same as $\varepsilon_1 = 3.8$ (quartz) and $\varepsilon_2 = 1.0$ (air), while the thicknesses are varied as $(t_1, t_2) = (1.00, 3.30)$, $(1.00, 3.60)$, and $(1.00, 3.90)$ mm, corresponding to the filling ratios $t_1/\Lambda = 23.26\%$, $21.74\%$, and $20.41\%$ for the periods of the stacks $\Lambda = 4.3$, $4.6$, and $4.9$ mm, respectively. As one can see, the central frequencies of the PBGs for both the $TE_{10}$ and $TM_{11}$ modes, as shown in Figure 2a,b respectively, are shifted to lower values as the filling ratio decreases. As mentioned above, the frequency axes in the plots are normalized to the cutoff frequency of $TE_{10}$ for general use.

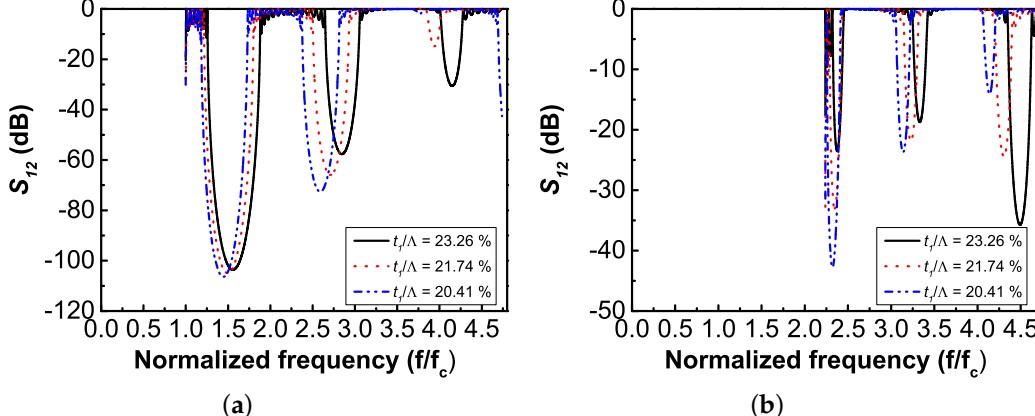

**Figure 2.** Calculated transmittance, $S_{12}$(dB), of the quasi-1D PCs with 15 double-layer stacks enclosed in the rectangular waveguide for (**a**) $TE_{10}$ and (**b**) $TM_{11}$ modes of light propagation. The dielectric constants used are $\varepsilon_1 = 3.8$ and $\varepsilon_2 = 1.0$, and the corresponding filling ratios are $t_1/\Lambda = 23.26\%$, $21.74\%$, and $20.41\%$ in the three cases. The frequency axes are normalized to the cutoff frequency of $TE_{10}$.

In order to look for a complete PBG for all possible modes in the quasi-1D PCs, one may want to enlarge a PBG for a specific mode of choice. For the following four cases, the dielectric constants used are $(\varepsilon_1, \varepsilon_2) = (2.3, 1.0)$, $(3.8, 1.0)$, $(4.9, 1.0)$, and $(11.4, 1.0)$, while the thicknesses are varied as $(t_1, t_2) = (2.15, 2.15)$, $(1.00, 3.30)$, $(0.72, 3.58)$, and $(0.27, 4.03)$ mm, corresponding to the filling ratios $t_1/\Lambda = 50.00\%$, $23.26\%$, $16.74\%$, and $6.28\%$ for the fixed period of the stacks $\Lambda = 4.3$ mm, respectively. The dielectric constants of 2.3, 3.8, 4.9 and 11.4 used in the chosen microwave frequency range correspond to the dielectric materials, polyethylene, quartz, phenolic resin, and barium sulfate, respectively. As one can see, the calculated transmittances, $S_{12}$(dB), of the four quasi-1D PCs with different 15 double-layer stacks for the $TE_{10}$ and $TM_{11}$ modes are plotted in Figure 3a,b, respectively. The width of PBGs for both the $TE_{10}$ and $TM_{11}$ modes in the quasi-1D PCs is widened to larger sizes as decreasing the filling ratio while increasing the dielectric contrast between $\varepsilon_1$ and $\varepsilon_2$. One may notice that the configurations of these quasi-1D PCs have been specially arranged so that the central frequencies of the PBGs of the $TE_{10}$ mode are kept the same while tuning the PBG sizes. However, those of the PBGs of the $TM_{11}$ mode are not centralized. This indicates that a tremendous computational effort is still needed to find a complete PBG in spite of the approach with a closed form developed is very efficient for the quasi-1D PCs.

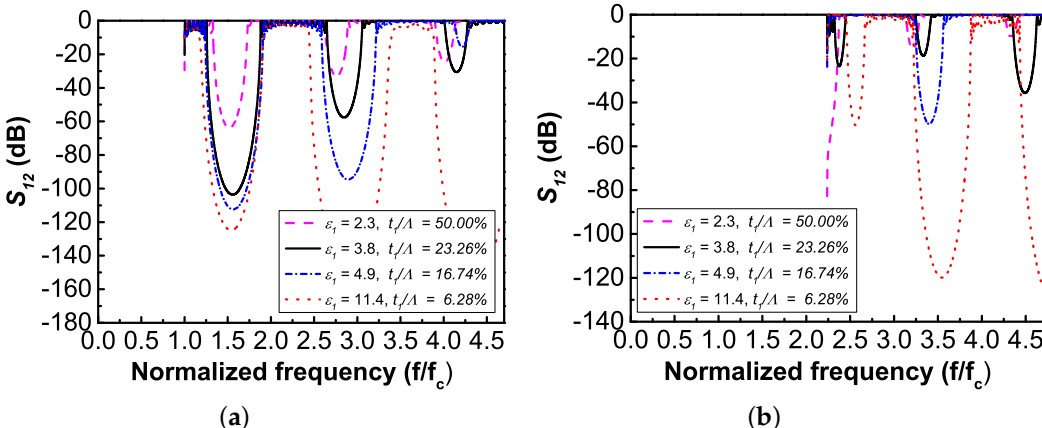

**Figure 3.** Calculated transmittance, $S_{12}$(dB), of the quasi-1D PCs with 15 double-layer stacks enclosed in the rectangular waveguide for (**a**) $TE_{10}$ and (**b**) $TM_{11}$ modes of light propagation. The dielectric constants used are $\varepsilon_1$ = 2.3, 3.8, 4.9, and 11.4 and $\varepsilon_2 = 1$, and the corresponding filling ratios are $t_1/\Lambda = 50.00\%$, 23.26%, 16.74%, and 6.28% in the four cases. The frequency axes are normalized to the cutoff frequency of $TE_{10}$.

A better way to identify PBGs more accurately and efficiently is to perform the BS calculation and further compute the PDOS [15,16]. Figure 4 shows the band structures of (a) $TE_{10}$ and (b) $TM_{11}$ modes for the quasi-1D PC with the dielectric constants $\varepsilon_1 = 3.8$, $\varepsilon_2 = 1$ and the thicknesses $t_1 = 1.00$ mm, $t_2 = 3.30$ mm, corresponding to the case presented in Figures 2 and 3 in a black solid line with a filling ratio of 23.26%. The PBGs of the $TE_{10}$ mode show no overlap with those of $TM_{11}$ mode, all marked in gray stripes in the figure. However, from Figure 4, one may notice that there are no photon states in the frequency ranges below the cutoff frequency of the $TE_{10}$ mode nor within the first PBG of $TE_{10}$ which is under the cutoff frequency of the $TM_{11}$ mode. Note that only the lowest TE amd TM modes are plotted in Figure 4. As there might be other modes involved within the frequency range of the PBG of interest, it is easier to identify a complete PBG from the PDOS plots compared to the BS ones. Figure 5 shows the PDOS of $TE_{10}$, $TE_{01}$, $TE_{11}$, and $TM_{11}$ modes for the same quasi-1D PC. As one can see, the PDOS of the $TE_{10}$ and $TM_{11}$ modes are consistent with the BS calculations shown in Figure 4. The PDOS contributed from the $TE_{01}$ and $TE_{11}$ modes tend to fill up the first PBG of the $TE_{10}$ mode. Other higher order modes are not considered as their cutoff frequencies are too high to contribute any PDOS in the frequency range of the PBG. Finally, the combined PDOS of $TE_{10}$, $TE_{01}$, $TE_{11}$, and $TM_{11}$ modes shows no photon states in some frequency ranges. The first one is below the cutoff frequency of the $TE_{10}$ mode, $(0–0.77)f_c$, the second one is within the first PBG of $TE_{10}$ but under the cutoff frequency of the $TE_{01}$ mode, $(1.26–1.48)f_c$, and the third one is the overlap of the PBGs of $TE_{10}$, $TE_{01}$, and $TE_{11}$ modes, $(1.79–1.87)f_c$. Therefore, a "complete PBG" can be obtained for some frequency ranges and categorized into three types: (1) below the cutoff frequency of the fundamental TE mode, (2) within the PBG of the fundamental TE mode but below the cutoff frequency of the next higher order mode, and (3) within an overlap of the PBGs of either TE modes, TM modes, or both.

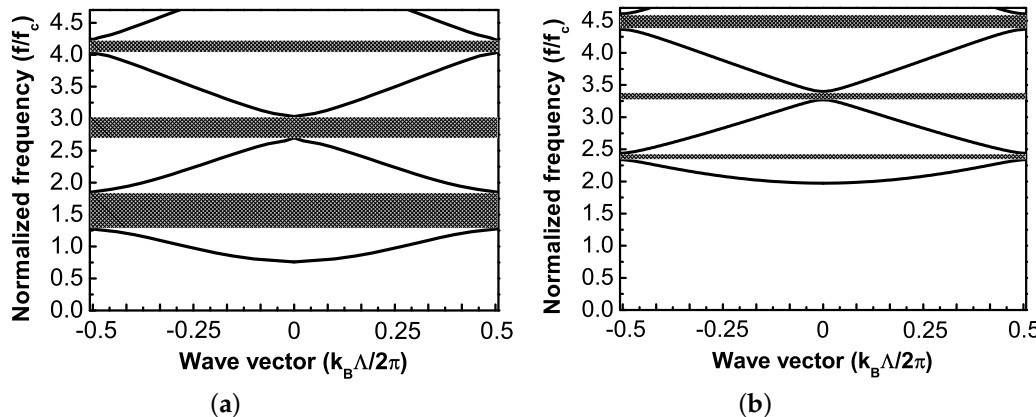

**Figure 4.** Band structures of (**a**) $TE_{10}$ and (**b**) $TM_{11}$ modes for the quasi-1D PC with the dielectric constants $\varepsilon_1 = 3.8$, $\varepsilon_2 = 1$ and the thicknesses $t_1 = 1.00$ mm, $t_2 = 3.30$ mm, corresponding to a filling ratio of 23.26%. The PBGs of $TE_{10}$ mode show no overlap with those of $TM_{11}$ mode.

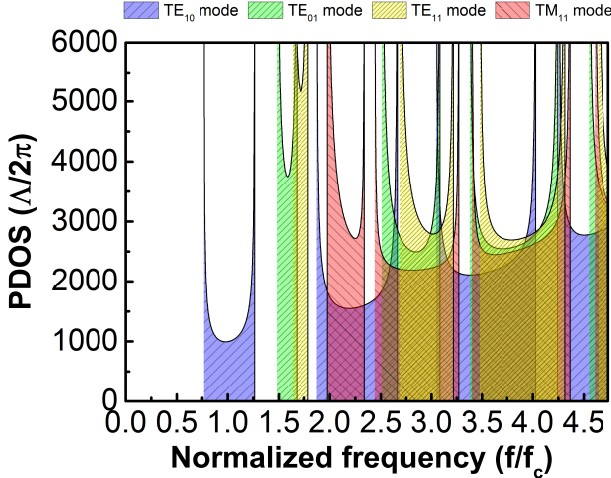

**Figure 5.** PDOS of $TE_{10}$, $TE_{01}$, $TE_{11}$, and $TM_{11}$ modes for the quasi-1D PC with the dielectric constants $\varepsilon_1 = 3.8$, $\varepsilon_2 = 1$ and the thicknesses $t_1 = 1.00$ mm, $t_2 = 3.30$ mm, corresponding to a filling ratio of 23.26%. The combined PDOS of $TE_{10}$, $TE_{01}$, $TE_{11}$, and $TM_{11}$ modes shows no photon states in some frequency ranges.

## 4. Conclusions

In summary, light propagation in quasi-1D PCs has been investigated quantitatively. The transmittances for both the TE and TM modes through a periodic multilayer heterostructure in a rectangular waveguide are calculated using the transfer matrix method. The corresponding band structures are obtained by solving the eigenvalue equations with proper periodic boundary conditions following the Bloch theorem. The formulas for determining the PDOS have been obtained to facilitate identifying the photonic band gaps for all the modes residing in the system. With our approach, the quantitative determination of PDOS can be performed very accurately and efficiently. A complete PBG can exist in these quasi-1D PCs, but the determination must be carefully conducted and verified. It is demonstrated that three types of "complete PBG" can be found and categorized as follows. The first type is the frequency range within which the TE and TM modes are both cutoff, the second type is for which the fundamental TE mode has a PBG while other higher order modes are cutoff, and the last type is an overlap of the PBGs of either TE modes, TM modes, or both. The model might be easier for an experimental validation in a millimeter wave frequency range while the optical counterpart might

be possibly pursued as well. We believe these results are of general importance and relevance to the dipole radiation or spontaneous emission by an atom in quasi-1D periodic structures and may have applications in future photonic quantum technologies.

**Author Contributions:** Conceptualization, M.-C.L.; methodology, R.-F.J. and M.-C.L.; software, R.-F.J. and M.-C.L.; validation, R.-F.J. and M.-C.L.; formal analysis, R.-F.J.; investigation, R.-F.J. and M.-C.L.; resources, R.-F.J. and M.-C.L.; data curation, R.-F.J.; writing—original draft preparation, R.-F.J. and M.-C.L.; writing—review and editing, R.-F.J. and M.-C.L.; visualization, R.-F.J.; supervision, M.-C.L.; project administration, M.-C.L.; funding acquisition, R.-F.J. and M.-C.L.

**Funding:** This work was partially supported by Guangdong Industry Polytechnic, P. R. China, under Grant Nos. KYRC2018-001 and RC2016-005, the research fund of Hanyang University (HY- 201400000002393), National Research Foundation of South Korea (2015R1D1A1A01061017), and the Alexander von Humboldt Foundation of Germany.

**Acknowledgments:** The authors would like to thank the late B. Y. Gu at the Institute of Physics, CAS and C. T. Chan at the Department of Physics, HKUST for the helpful comments, discussions, and encouragement.

**Conflicts of Interest:** The authors declare no conflict of interest. The funders had no role in the design of the study; in the collection, analyses, or interpretation of data; in the writing of the manuscript, or in the decision to publish the results.

## Abbreviations

The following abbreviations are used in this manuscript:

| | |
|---|---|
| 1D | one-dimensional |
| 2D | two-dimensional |
| 3D | three-dimensional |
| BS | band structure |
| EM | electromagnetic |
| PBG | photonic band gap |
| PC | photonic crystal |
| PDOS | photon density of states |
| TE | transverse electric |
| TM | transverse magnetic |

## Appendix A. FORMULAS

Here, we give the formulas of the functions employed in Equations (41) and (42).

$$\alpha_1 = \exp[-j(k_1t_1 + k_2t_2)]\left[\exp(j2k_1t_1) + \exp(j2k_2t_2)\right]$$
$$\left[\frac{\omega}{k_1}\left(\frac{\varepsilon_1\mu_1k_2}{k_1^2} - \frac{\varepsilon_2\mu_2}{k_2}\right)\right]\left(-\frac{k_1}{k_2} + \frac{\mu_1}{\mu_2}\right), \tag{A1}$$

$$\alpha_2 = \exp[-j(k_1t_1 + k_2t_2)]\{1 + \exp[j2(k_1t_1 + k_2t_2)]\}$$
$$\left[\frac{\omega}{k_1}\left(-\frac{\varepsilon_1\mu_1k_2}{k_1^2} + \frac{\varepsilon_2\mu_2}{k_2}\right)\right]\left(\frac{k_1}{k_2} + \frac{\mu_1}{\mu_2}\right), \tag{A2}$$

$$\alpha_3 = \exp[-j(k_1t_1 + k_2t_2)]\left[\exp(j2k_1t_1) + \exp(j2k_2t_2)\right]$$
$$\left[\frac{\omega}{k_2}\left(-\frac{\varepsilon_1\mu_1}{k_1} + \frac{\varepsilon_2\mu_2k_1}{k_2^2}\right)\right]\left(-\frac{k_2}{k_1} + \frac{\mu_2}{\mu_1}\right), \tag{A3}$$

$$\alpha_4 = -j\exp[-j(k_1t_1 + k_2t_2)]\left[\exp(j2k_1t_1) - \exp(j2k_2t_2)\right]$$
$$\left[\omega\left(-\frac{\varepsilon_1\mu_1t_1}{k_1} + \frac{\varepsilon_2\mu_2t_2}{k_2}\right)\right]\left(-\frac{k_1}{k_2} + \frac{\mu_1}{\mu_2}\right)\left(\frac{k_2}{k_1} + \frac{\mu_2}{\mu_1}\right), \tag{A4}$$

$$\alpha_5 = \exp[-j(k_1 t_1 + k_2 t_2)]\{1 + \exp[j2(k_1 t_1 + k_2 t_2)]\}$$
$$\left[\frac{\omega}{k_2}\left(\frac{\varepsilon_1 \mu_1}{k_1} - \frac{\varepsilon_2 \mu_2 k_1}{k_2^2}\right)\right]\left(\frac{k_2}{k_1} + \frac{\mu_2}{\mu_1}\right), \tag{A5}$$

$$\alpha_6 = -j\exp[-j(k_1 t_1 + k_2 t_2)]\{-1 + \exp[j2(k_1 t_1 + k_2 t_2)]\}$$
$$\left[\omega\left(-\frac{\varepsilon_1 \mu_1 t_1}{k_1} - \frac{\varepsilon_2 \mu_2 t_2}{k_2}\right)\right]\left(\frac{k_1}{k_2} + \frac{\mu_1}{\mu_2}\right)\left(\frac{k_2}{k_1} + \frac{\mu_2}{\mu_1}\right), \tag{A6}$$

$$\beta_1 = \exp[-j(k_1 t_1 + k_2 t_2)]\left[\exp(j2k_1 t_1) + \exp(j2k_2 t_2)\right]$$
$$\left[\frac{\omega}{k_1}\left(\frac{\varepsilon_1 \mu_1 k_2}{k_1^2} - \frac{\varepsilon_2 \mu_2}{k_2}\right)\right]\left(-\frac{k_1}{k_2} + \frac{\varepsilon_1}{\varepsilon_2}\right), \tag{A7}$$

$$\beta_2 = \exp[-j(k_1 t_1 + k_2 t_2)]\{1 + \exp[j2(k_1 t_1 + k_2 t_2)]\}$$
$$\left[\frac{\omega}{k_1}\left(-\frac{\varepsilon_1 \mu_1 k_2}{k_1^2} + \frac{\varepsilon_2 \mu_2}{k_2}\right)\right]\left(\frac{k_1}{k_2} + \frac{\varepsilon_1}{\varepsilon_2}\right), \tag{A8}$$

$$\beta_3 = \exp[-j(k_1 t_1 + k_2 t_2)]\left[exp(j2k_1 t_1) + exp(j2k_2 t_2)\right]$$
$$\left[\frac{\omega}{k_2}\left(-\frac{\varepsilon_1 \mu_1}{k_1} + \frac{\varepsilon_2 \mu_2 k_1}{k_2^2}\right)\right]\left(-\frac{k_2}{k_1} + \frac{\varepsilon_2}{\varepsilon_1}\right), \tag{A9}$$

$$\beta_4 = -j\exp[-j(k_1 t_1 + k_2 t_2)]\left[\exp(j2k_1 t_1) - \exp(j2k_2 t_2)\right]$$
$$\left[\omega\left(-\frac{\varepsilon_1 \mu_1 t_1}{k_1} + \frac{\varepsilon_2 \mu_2 t_2}{k_2}\right)\right]\left(-\frac{k_1}{k_2} + \frac{\varepsilon_1}{\varepsilon_2}\right)\left(\frac{k_2}{k_1} + \frac{\varepsilon_2}{\varepsilon_1}\right), \tag{A10}$$

$$\beta_5 = \exp[-j(k_1 t_1 + k_2 t_2)]\{1 + \exp[j2(k_1 t_1 + k_2 t_2)]\}$$
$$\left[\frac{\omega}{k_2}\left(\frac{\varepsilon_1 \mu_1}{k_1} - \frac{\varepsilon_2 \mu_2 k_1}{k_2^2}\right)\right]\left(\frac{k_2}{k_1} + \frac{\varepsilon_2}{\varepsilon_1}\right), \tag{A11}$$

and

$$\beta_6 = -j\exp[-j(k_1 t_1 + k_2 t_2)]\{-1 + \exp[j2(k_1 t_1 + k_2 t_2)]\}$$
$$\left[\omega\left(-\frac{\varepsilon_1 \mu_1 t_1}{k_1} - \frac{\varepsilon_2 \mu_2 t_2}{k_2}\right)\right]\left(\frac{k_1}{k_2} + \frac{\varepsilon_1}{\varepsilon_2}\right)\left(\frac{k_2}{k_1} + \frac{\varepsilon_2}{\varepsilon_1}\right). \tag{A12}$$

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
