# Peer review of "Quantitative Analysis of Photon Density of States for One-Dimensional Photonic Crystals in a Rectangular Waveguide"

_crystals, doi:10.3390/cryst9110576_

Round 1

Reviewer 1 Report

The manuscript ‘Quantitative Analysis of Photon Density of States for One-dimensional Photonic Crystals in a Rectangular Waveguide’ by Jao and Lin considers photonic density of states in one-dimensional periodic structure in a rectangular waveguide. The authors find overlapping of TM and TE gaps, which makes a complete photonic band gap.

I find that manuscript of some interest but it has major problems. The main one is that the authors inexplicitly use PEC walls conditions which unreachable in optical wavelength. At the same time authors claim their “results are of general importance and relevance to the spontaneous emission by an atom or to dipole radiation in quasi-1D periodic structures and might have applications in future photonic quantum technologies”. When I read the manuscript I could not understand what a problem is under study. I was really confused when found that it is a microwave problem, which is definitely incompatible with either spontaneous emission or quantum technologies. After reading the abstract I was sure that the authors apply something similar to the Marcatili’s method widely used for optical waveguides.

The second problem is that for analysis of radiation sources photonic density of states are not useful. I suggest authors to consider local density of states instead. Calculation methods for one-dimension systems are discussed in the following references:

Denning et al. arxiv 1907.13423 (2019);

Gregersen et al. Optics  Express24,  20904 (2016);

Rybin et al. Sci. Reports 6, 20599 (2016);

Denning, et al Physical  Review  B98,121306(R) (2018).

Overall, I cannot recommend publication of this manuscript. I suggest a major revision with clear message whether this paper is for microwaves and applications in that range or for optics with applications in spontaneous emission or quantum technologies.

Some minor issues:

- reference numbering: Refs. 9,10 before Ref. 8;

- what is ‘the number of lattices’?

- what is ‘a wide-band transmission of millimeter’? Is millimeter a bandwidth?

Reviewer 2 Report

The work has the value of extending the Transfer Matrix approach, commonly used for wave propagation through one-dimensional multilayered structures, to multilayered rectangular waveguides, thus including the mode information. Derivations of the close-form formulas are rigorous and sufficiently detailed. The manuscript is clear and the claims are well-supported by the results. However, some typos here and there should be checked.

I would ask the authors to explain how the dielectric constants used in the calculation have been chosen. Are they related to real materials commonly used in microwave devices?

Reviewer 3 Report

The paper “Quantitative Analysis of Photon Density of States for One-dimensional Photonic Crystals in a Rectangular Waveguide” by Jao et al presents the numerical formulation and quantitative analysis of one-dimensional photonic crystal embedded in a waveguide. The aim of the authors is to explore configurations with a complete photonic bandgap. The authors individuate, through transfer matrix method, a particular configuration of a simple mono-dimensional quasi-1D photonic crystal characterized by a “complete PBG” in specific frequency range. The story is well structured, although the authors should provide more insights on how this formulation could be potentially useful to other more complex structures or devices and at the same time specify on how this formulation could be considerably different than others currently employed, lowering the overall impact and usefulness of the community. In its current form I have some critical objections related to the technical and in parts to the presentation of results. I therefore suggest major revisions.

The concept of complete PBG is not introduced properly nor well-defined. I believe that would help the reader to understand more clearly the aim of the paper. Also, as con is sometimes hyphenated, in upper commas and other times not.

The two-layer unit cells periodic array investigated in this framework is rather simple and doesn’t present great novelty considering the well-known studies on waveguide bragg grating as a 1D photonic bandgap structure or easily solvable models with commercially available softwares such as Optiwave (Transfer matrix approach), Comsol (FEM) or CST (FDTD). Therefore, I would relegate the formulation, especially the part where the authors derive the  transversal Helmholtz’s equation for the waveguide, to supplementary information or appendix, and focus more on the investigation of the complete PBG and its potential applications.

In the introduction, I advise the author to provide specific citations not in the aggregate form. It would be useful, to refer to each field in which PC could be useful with specific citations. In details, to the following sentence “These PC based materials are expected to have many applications in optoelectronics, optical communications, and photonic quantum technologies in the forthcoming decades [5]. “ Also, the phrase:  “One the other hand, there has been a lot of interests in microwave and millimeter wave  […] such as the significant progress in the design of filters, microstrip antennas, slow wave structures, and backward-wave devices, and so on “ would require citations for each of the device typology rather than the end of the sentence.

In the introduction, for the sake of competition, the authors should refer to the more recent and very timely hyperbolic metamaterials such as Metal insulator metal dual periodic structures (for instance: Optica Vol. 3, No. 12; Optics Letters Vol. 39, Issue 16, pp. 4663-4666(2014), OPTICS LETTERS / Vol. 34, No. 16 / August 15, 2009) or planar 2-dimensional aperiodic crystals used for antennas and polarizers (to cite few: ACS Nano 2019 13 5 5646-5654; Optics Express Vol. 15, Issue 23, pp. 15314-15323 (2007); ACS Photonics2018562418-2425) in the optical domain.

In the numerical examples, I don’t have fully clear which are the criteria used for selecting  a specific PC configurations.  Authors should specify why they use those values of permittivity, filling factor and periodicity (i.e. material type and its availability). These parameters are essential for controlling the frequency response and the mode dispersion of the PC and should be explained with more precision, especially considering the sentence: “The model might be easier for  an experimental validation in a millimeter wave frequency range while the optical counterpart might be pursued as well.

What is the angle at which the elelctro-magnetic wave is out-coupled from the PC at different frequencies? This might be an important thing to investigate or discuss, conisdering the current trends in integrated LiDar technology solutions.

The authors don’t account for possible fabrication imperfection or irregularities which could be modelled as imperfection of the periodicity of the function epsilon(z). Authors consider materials for the dual-layer unit cell which are not affected by any losses (eps’’= 0). The formulation doesn’t consider the slope of the sidewalls which could pay a critical role. Would that affect the PBG? It should be interesting to add either one of these cases for providing more relation to a possible implementation but also to have better control over the photonic bandgap.

The impact of the study would sensitively improve if the phase of the periodic crystal (Filling Factor) is modulated according a function of the space. This function could be used for engineer and have more control over the the bandgap and photonic density of states for optimized configurations and specific target applications.

I would avoid refer to previous work directly in the abstract or in general any citation, which can be instead discussed in the introduction. Besisdes, avoiding  being autoreferential, it will help the story being more self-consistent within this article. 

Conclusions and discussion of the results are overly hasty. The authors only stated their numerical results, but didn’t compare their findings with others’ work, or interpret their own scientific understanding behind the results, or produce some further improvement plans, which based on the numerical results the refree believes there is ample margin to do that. In this iteration, the current discussion section looks more like a summary. 

Non-technical questions

Or otherwise is spelled vice versa not vise versa.

Qausi 1dphotonic crystal is misspelled and should be Quasi

Round 2

Reviewer 1 Report

The authors have failed to make major improvements of the manuscript. Both of the major problems raised by the reviewer were ignored.

 1) The difference of a metal response in the optical and the microwave ranges is not just in dissipations. Actually, I do not know a recipe to arrange PEC walls in the optical wavelengths. Unfortunately, even superconductors cannot help to solve this problem.

2) Photonic density of states is useful for an infinite system only. Authors consider the finite samples and have to analyze local density of states instead.

Moreover authors did not change ‘number of lattice’ (the number of lattice is one in their study) to something more appropriate.

I believe the manuscript have to be rejected.

Reviewer 3 Report

I appreciate the authors effort in modifying the paper moving towards the reviewers' suggestions. The paper has significantly improved its readability and authors solved some of my main concerns. 

I suggest the authors to consider focusing on a specific wavelength range (either microwave or optics) or they can still keep the formulation valid for EMW and photonic waveguides, but they would need to specify that boundary condition changed by simply adding a sentence for clearing out any other possible doubt from the readers. 

The authors should provide the material selected for the last example, which reflects the value of permittivity selected, I think it would help the manuscript being referred in design of similar chips.

I would avoid refer to photon states when dealing with microwave frequencies (especially in the example when the derivation loses generality) and probably use quotation marks. Also, when talking about microwave version of photonic crystal I would suggest to cite Yablonovitch's PRL paper

Round 3

Reviewer 1 Report

I find the authors disagree with my previous comments on the major problems. In the present way the manuscript has a narrow interest only. Indeed, the reported results are not surprising and the methods exploited in the manuscript are well-known. Actually I am giving transfer matrix problems as a part of practice for my master course on theoretical photonics. The second component of the present manuscript (the electromagnetic modes in rectangular waveguides) is from bachelor courses of electrodynamics.